# Neuroimmune Mechanisms Underlying Neuropathic Pain: The Potential Role of TNF-α-Necroptosis Pathway

**DOI:** 10.3390/ijms23137191

**Published:** 2022-06-28

**Authors:** Yi-Wen Duan, Shao-Xia Chen, Qiao-Yun Li, Ying Zang

**Affiliations:** 1Pain Research Center and Department of Physiology, Zhongshan Medical School, Sun Yat-sen University, 74 Zhongshan Road. 2, Guangzhou 510080, China; dyiwennn@163.com (Y.-W.D.); liqy223@mail2.sysu.edu.cn (Q.-Y.L.); 2Department of Anesthesiology, Sun Yat-sen University Cancer Center, State Key Laboratory of Oncology in South China, Collaborative Innovation Center for Cancer Medicine, Guangzhou 510060, China; chenshx1@sysucc.org.cn

**Keywords:** necroptosis, neuroinflammation, neuropathic pain, tumor necrosis factor-alpha (TNF-α)

## Abstract

The neuroimmune mechanism underlying neuropathic pain has been extensively studied. Tumor necrosis factor-alpha (TNF-α), a key pro-inflammatory cytokine that drives cytokine storm and stimulates a cascade of other cytokines in pain-related pathways, induces and modulates neuropathic pain by facilitating peripheral (primary afferents) and central (spinal cord) sensitization. Functionally, TNF-α controls the balance between cell survival and death by inducing an inflammatory response and two programmed cell death mechanisms (apoptosis and necroptosis). Necroptosis, a novel form of programmed cell death, is receiving increasing attraction and may trigger neuroinflammation to promote neuropathic pain. Chronic pain is often accompanied by adverse pain-associated emotional reactions and cognitive disorders. Overproduction of TNF-α in supraspinal structures such as the anterior cingulate cortex (ACC) and hippocampus plays an important role in pain-associated emotional disorders and memory deficits and also participates in the modulation of pain transduction. At present, studies reporting on the role of the TNF-α–necroptosis pathway in pain-related disorders are lacking. This review indicates the important research prospects of this pathway in pain modulation based on its role in anxiety, depression and memory deficits associated with other neurodegenerative diseases. In addition, we have summarized studies related to the underlying mechanisms of neuropathic pain mediated by TNF-α and discussed the role of the TNF-α–necroptosis pathway in detail, which may represent an avenue for future therapeutic intervention.

## 1. Introduction

Massive unregulated release of pro-inflammatory cytokines (cytokine storm) such as tumor necrosis factor-alpha (TNF-α) and interleukin-6 (IL-6) due to infection, nerve injury or immunotherapy causes organ damage and unbearable pain [1]. Neuropathic pain, caused by inflammatory cytokines, is a prominent symptom of cytokine storm and is characterized by a burning or electrical-like sensation, an unpleasant somatosensory experience leading to hyperalgesia (increased sensitivity to noxious stimuli), allodynia (pain evoked by normally innocuous stimuli) and spontaneous pain [2,3,4,5,6,7,8,9,10,11]. It is mainly secondary to central or peripheral nerve damage, cancer, diabetes, infection, stroke or immune and inflammatory disorders [12,13,14,15]. Because neuropathic pain is complex, impairs the quality of life of patients, increases the economic burden of patients [16,17,18,19] and lacks effective therapies, it is a clinical dilemma that warrants immediate attention. Over the past 30 years, various experimental and animal models have been used to examine the underlying mechanisms of neuropathic pain; among which, neuroimmune mechanisms have received increasing attention [20,21,22,23,24,25,26,27,28]. Although neuropathic pain is a sensory disorder, in the lumbar 5 ventral root transection (L5-VRT) model, motor fiber injury but not sensory fiber injury is crucial to the onset and development of neuropathic pain because it initiates neuroinflammatory responses such as the elevation of pro-inflammatory cytokine levels in pain-related pathways, thus facilitating peripheral and central sensitization [21,22,23,24,29,30,31,32,33,34]. Intra-sciatic injection or peri-sciatic administration of exogenous tumor necrosis factor-alpha (TNF-α) without any nerve injury reproduces pain hypersensitivity similar to that of neuropathic pain in humans [35,36], indicating that neuroinflammatory responses, instead of nerve injury, are necessary and sufficient for inducing neuropathic pain.

Microglia in the healthy brain continuously palpate the surrounding tissue for subtle disturbances [37] and can rapidly respond to tissue injury by altering morphological characteristics, proliferating and expressing a wide variety of inflammatory cytokines and chemokines [38]. Multiple studies have indicated that the local microenvironment plays an important role in regulating the microglial phenotype [39,40,41,42,43,44]. Microglial activation can be initiated by injured neurons [45,46,47,48], which contributes to central nervous system (CNS) pathology, such as in models of neuropathic pain [22,46,48,49,50,51,52]. A recent study showed that impaired death (necroptosis) and/or repopulation of microglia underpin their dysregulated activation in neurological diseases [53]; in addition, necroptosis can be triggered by many death receptors, including Fas, TRAIL and TNF receptors, mainly through the signalling pathway induced via the binding of TNF-α to TNF receptor 1 (TNFR1) [54]. Therefore, the TNF-α/TNFR–necroptosis pathway, a biochemical pathway causing programmed neurodegeneration and/or microglia death, may be responsible for or contribute to neuropathic pain. This review focuses on the peripheral and central mechanisms of neuropathic pain mediated by TNF-α, provides data regarding available potential therapeutic targets and suggests future research directions related to the TNF-α/TNFR–necroptosis pathway for neuropathic pain.

## 2. Inflammatory Response, Apoptosis and Necroptosis Induced by TNF-α

TNF-α is important for mammalian immunity and cellular homeostasis. The role of TNF-α as a master regulator in balancing cell survival and death has been extensively studied in various cell types and tissues. As shown in Figure 1, TNF-α induces an inflammatory response and two programmed cell death mechanisms, namely, apoptosis and necroptosis, based on different pathological conditions [54,55,56,57,58]. Receptor interacting protein kinase 1 (RIPK1) and TNF receptor-associated death domain (TRADD) regulate TNF-dependent signalling, which controls the balance between cell death and survival [59].

### 2.1. Cell Survival and Inflammatory Response

When TNF-α binds to TNFR1 on the membrane surface, the conformation of TNFR1 changes, and TNFR1 complex I is rapidly formed via the recruitment of various proteins, including TRADD, RIPK1, TNF receptor-associated factor 2 (TRAF2) and cellular inhibitor of apoptosis protein 1 and 2 (cIAP1/2). TRADD is very important for the recruitment of TRAF2 and ubiquitination of RIPK1 and complex I. The TGF-activated kinase 1 (TAK1) binding protein (TAB) complex and IκB kinase (IKK) complex consisting of IKK1 can prevent cell death, whereas cIAP1/2 can prevent TNFR1-mediatednecroptosis and promote ubiquitination of RIPK1 [60,61]. Ubiquitination of RIPK1 activates the NF-κB signalling pathway through TAK and IKK complexes, promotes cell survival and induces an inflammatory response [62].

### 2.2. Apoptosis

Deubiquitination of RIPK1 results in the formation of either complex IIa or complex IIb. If caspase-8 is present in cells, TNFR1 complex I recruits TRADD and caspase-8 to form complex IIa (composed of TRADD, FADD and caspase-8), activating a caspase cascade and leading to RIPK1-independent apoptosis (RIA). When receptor-interacting protein kinase 3 (RIPK3) and mixed lineage kinase domain-like (MLKL) protein are fully expressed and caspase-8 is present, ubiquitination of RIPK1 is inhibited, and it interacts with RIPK3 to form complex IIb (mainly composed of FADD, caspase-8, RIPK1 and RIPK3), which activates caspase-8 and triggers RIPK1-dependent apoptosis (RDA) [63,64].

### 2.3. Necroptosis

Stimulation of the Fas/TNFR family can not only trigger a canonical ‘extrinsic’ apoptotic pathway but also activate necroptosis inhibited by necrostatin-1 (Nec-1), a specific and potent small-molecule substance, in the absence of intracellular apoptotic signalling [65]. When caspase-8 is inactive or inhibited, necroptosis is initiated via complex IIb. RIPK1 recruits RIPK3 and induces auto- and trans-phosphorylation, with consequent oligomerisation of the phosphorylated RIPK3. After phosphorylation, RIPK1 and RIPK3 form a necrosome (a multiprotein complex resembling amyloids) with MLKL. RIPK3 recruits MLKL and phosphorylates it. Subsequently, MLKL oligomerises and migrates to the cell membrane from the cytoplasm, which results in membrane permeabilization owing to the binding of MLKL to phosphatidylinositol lipids and cardiolipin, thus leading to cell death [66,67].

## 3. Role of TNF-α and Its Mechanism Underlying Neuropathic Pain

TNF-α, which primarily activates a cascade of other cytokines [68], is upregulated in both the peripheral nervous system [21,69,70,71] and CNS, including the hippocampus [72,73], locus coeruleus [72,74], medial prefrontal cortex (mPFC) [75], anterior cingulate cortex(ACC) [20] and spinal cord [21] in animal pain models, thereby mediating the initiation and maintenance of neuropathic pain (Figure 2) [66,76,77,78].

### 3.1. TNF-α Regulates Voltage-Gated Sodium Channels in the Peripheral Nervous System

Studies have shown that TNF-α facilitates heat-induced CGRP release from nociceptor terminals in the skin [79] and sensitizes peripheral Aβ- and C-fibers [80,81,82,83]. This sensitization of peripheral primary afferents by TNF-α results from the abnormal expression of voltage-gated sodium channels (VGSCs) in the dorsal root ganglia (DRG) [84,85,86]. Selective injury of motor fibers via L5-VRT upregulates the expression of TNF-α [21], tetrodotoxin-sensitive (TTX-S) Nav1.3 and tetrodotoxin-resistant (TTX-R) Nav1.8 [23] and increases the current density of TTX-S and TTX-R channels in DRG [23,87]. The increase in the expression of the above mentioned VGSCs and sodium currents is reported to be significantly lower in TNFR1-knockout mice than in wild-type mice [23,87]. Furthermore, peri-sciatic exogenous TNF-α, which leads to lasting mechanical allodynia [71], upregulates Nav1.3 and Nav1.8 in DRG in vivo. In addition, TNF-α-induced Nav1.3 and Nav1.8 can be formed in cultured adult rat DRG in a dose-dependent manner [23]. Nuclear factor-kappaB (NF-κB) and p38 mitogen-activated protein kinase (p38 MAPK) pathways mediate the abnormal expression of VGSCs or sodium currents following nerve injury [25,87].

Nav1.6 (*SCN8A*), another major TTX-S VGSC in the mammalian nervous system, regulates neuronal activity at the axon initial segment [88], relays excitatory persistent and resurgent currents in DRG [89] and is involved in the production and maintenance of pathological neuronal excitability in the peripheral nerves [90,91]. In DRG, spontaneously active bursting cells express high levels of Nav1.6 and knockdown of Nav1.6 completely blocks the abnormal spontaneous activity and pain behaviors [84,90,91,92,93]. TNF-α can directly regulate the expression and function of Nav1.6 by epigenetically upregulating Nav1.6 expression via the signal transducer and activator of transcription-3 (STAT3) pathway, which promotes the trafficking of Nav1.6 to the membrane of neurons in DRG, an essential step in mediating neuronal excitability and repetitive firing, thus contributing to neuropathic pain [84,94].

TTX-S Nav1.7 (*SCN9A*) has been intensively studied in the sensory system. In humans, the loss of function of Nav1.7 leads to a complete inability to sense pain [95,96], and mutations inNav1.7 lead to the functional absence of nociceptors [97]. However, the gain of function of Nav1.7 results in paroxysmal extreme pain disorder [98]. In rodents, deletion or blockage of Nav1.7 in mouse DRG attenuates nerve injury-, inflammation-, burn injury- and paclitaxel (a chemotherapeutic drug)-induced chronic pain [99,100,101,102,103]. Previous studies have shown that TNF-α/NF-κB signalling induces hypersensitivity in DRG via nuclear transcription [104]. In addition, a recent study showed that NF-κB *p*-p65 non-transcriptionally gates Nav1.7 channels in the membrane of neurons in rat DRG, and TNF-α contributes to the protein–protein interaction between *p*-p65 and Nav1.7 within several minutes [105], revealing the possible mechanism underlying the rapid regulation of sodium currents by TNF-α [87].

In addition to regulating VGSCs, TNF-α can increase the conductance of transient receptor potential ankyrin 1 (TRPA1) [106], transient receptor potential vanilloid 1(TRPV1) [107,108], voltage-gated calcium channel subunit CaV3.2 [86] and membrane K^+^ ions [109]. In response to nerve injury and inflammation, overexpression of TNF-α in the peripheral nervous system sensitises primary afferents by enhancing the cation channels to induce neuropathic pain.

### 3.2. TNF-α Induces Spinal Neuronal Excitation and Inhibition Imbalance Andneuroinflammation

In the spinal cord, TNF-α enhances excitatory synaptic transmission and increases AMPA- and NMDA-evoked excitatory postsynaptic currents (EPSCs) to induce pain hypersensitivity [110]. After nerve injury, the upregulation of TNF-α may activate NF-κB, p38 MAPK and JNK via TNFR1 on neurons and glial cells to induce long-term potentiation (LTP) of C-fibre-evoked field potentials in the spinal dorsal horn [111]. In addition, TNF-α can reduce inhibitory synaptic transmission, indicating that disinhibition of synaptic transmission is mediated by TNF-α/TNFR1 followed by the activation of the p38 MAPK pathway in GABAergic neurons of the spinal cord [78]. In addition to the neuronal excitation and inhibition imbalance, TNF-α-induced neuroinflammation and spinal microglial activation contribute to neuropathic pain by initiating the release of other proinflammatory cytokines and promoting crosstalk between neurons and glial cells to affect synaptic signalling and pain transmission [21,22,112]. In the spinal cord, activation of TNF-α after spinal cord injury and noxious stimulation promotes NF-κB, extracellular signal-regulated kinase (ERK), JNK and caspase-8 pathways, thereby initiating inflammatory and apoptotic processes to affect the development and maintenance of neuropathic pain [113]. Furthermore, microglia-derived TNF-α elevates the expression of cyclooxygenase 2 and prostaglandin I2 (PGI2) synthase in spinal endothelial cells, which promotes neuropathic pain via the neuronal PGI2 receptor [114], suggesting that the glial–endothelial cell interaction of the neurovascular unit via transient TNF-α is responsible for the generation of neuropathic pain.

### 3.3. Supraspinal TNF-α Mediates Neuropathic Pain, Pain-Associated Aversion, Anxiety, Depression and Memory Deficits

Changes in higher-order functions, such as learning and memory disorders, anxiety or depression, are critical components of pain phenotypes, especially in a chronic pain state [115,116,117,118]. TNF-α plays a key role in supraspinal modulation of pain transduction; for example, intracerebroventricular (ICV) injection of TNF-α induces hyperalgesia [119,120]. In addition, blocking TNF-α in the brain reduces neuropathic pain, pain-associated aversion and memory deficits [20,117,121]. Glial cells are considered a major source of cytokines and chemokines in the brain [122], and the activation of microglia in ACC, hippocampus, prefrontal cortex and other brain regions associated with pain information processing contributes to pain aversion [123], memory deficits [124], anxiety and depression [124] and may be associated with overexpression of TNF-α [20,125]. Reciprocal activation between neurons and microglia facilitates pain transmission [75,121,126].

At the molecular level, supraspinal TNF-α modulates excitatory and inhibitory synaptic transmission in different ways [127]. Overproduction of TNF-α enhances the phosphorylation of NMDA receptor subunit NR1 in the rostral ventromedial medulla (RVM), contributing to descending pain facilitation [121]. In oxaliplatin-induced pain models, TNF-α is upregulated in the dorsolateral region of the midbrain periaqueductal gray, which is accompanied by an impaired GABAergic descending inhibitory system [128]. Activation of the NF-kB, ERK, p38 MAPK and JNK pathways induced by supraspinal TNF-α is responsible for SNI-induced neuropathic pain [129]. Furthermore, activation of TNF-α/TNFR1 and microglia following nerve injury mediates opposite structural synaptic alterations in spinal (induction of LTP) [111,112] and hippocampal (inhibition of LTP) neurons [130,131,132,133,134,135], which may be used to explain hyperalgesia/allodynia and pain-associated memory disorders, respectively.

## 4. TNF-α/TNFR1-Necroptosis Pathway in Neuropathic Pain

In addition to TNF-α-mediated neuroinflammation and excitotoxicity [20], apoptosis induced by TNF-α/TNFR1 [64,136,137,138] also promotes peripheral neuropathic pain [139,140]. Necroptosis, a novel form of TNF-α-mediated cell death, has attracted increasing attention in the past decade. As shown in Figure 3A, the number of articles published worldwide shows a steady upward trend, with the research explosion point of necroptosis in the field of neuroscience beginning in 2014.

### 4.1. Necroptosis and Bibliometric Analysis 

Necroptosis is characterised by necrotic cell death and autophagy activation [65,146]. In 1988, Laser et al. found that cell necrosis can be regulated and occurs actively [141]. Subsequently, some studies reported a death mechanism that lacked apoptotic signalling; however, the morphological features of cell death were similar to those of necrosis [142]. After a few years, Chan et al. reported receptor-interacting protein (RIP)-dependent programmed necrosis [143]. In addition, Degterev et al. found that Nec-1 and related molecules can regulate the aforementioned form of cell death and termed it necroptosis [65,144]. Finally, necroptosis was officially named in 2018 [145]. The developmental process is shown in Figure 3B. Necroptosis results in morphological characteristics similar to those of necrosis and activates autophagy but appears to be tightly regulated [65,147,148].

The literature published on necroptosis in the field of neuroscience from 2012 to 2021 was quantitatively and qualitatively analyzed using the Bibliometrix R Package and VOSviewer software [149,150], and the top 10 articles with the highest co-citation rate and the top 10 most cited articles are listed in Table 1, Table 2 and Table 3, respectively. Most publications in Table 1 belong to Q1 in the Journal Citation Reports (JCR) division, reflecting the necessity of in-depth research on necroptosis to a certain extent. Research into necroptosis in the field of neuroscience has gradually deepened, with studies focusing on various aspects from molecular mechanisms to the pathogenesis of central diseases (Table 2 and Table 3).

Keyword co-occurrence analysis performed using VOSviewer revealed four categories of keywords related to necroptosis, which are shown in red, blue, green and yellow in Figure 4. The red clusters are largest, and ‘necroptosis’ constitutes the largest node. In addition, the occurrence of terms such as ‘inflammation’, ‘oxidative stress’, ‘activation’ and ‘neurodegeneration’ suggests that necroptosis activation is closely related to neuroinflammation and neurodegeneration.

### 4.2. TNF-α/TNFR1–Necroptosis Pathway Contributes to Neurological Diseases

As mentioned above (Figure 1), necroptosis is activated by ligands of death receptors such as TNF-α under caspase-deficient conditions. RIPK3 and MLKL are important for necroptosis activation [58,152,153,154,155,158,165,166]. TNF-α/TNFR1-induced necroptosis occurs in several neurodegenerative diseases of the CNS, such as multiple sclerosis (MS), Parkinson’s disease and Alzheimer’s disease [156,167,168,169,170]. After an ischaemic stroke, perivascular microglia-induced endothelial necroptosis leading to disruption of the blood–brain barrier requires TNF-α to act on its receptor TNFR1 [57]. Furthermore, chronic pain induced by SNI sensitizes the heart to myocardial ischaemia–reperfusion injury, and myocardial necroptosis plays an important role in this pathophysiological process, with an increase in TNF-α levels following by a robust interaction between RIP1/RIP3 and RIP3-induced phosphor-MLKL/CaMKII signalling [171].

### 4.3. Role of Necroptosis in Chronic Pain

Necroptosis, a mode of programmed cell death similar to necrosis and conventional apoptosis, is usually accompanied by plasma membrane rupture, organelle swelling and inflammatory cell infiltration [61,147]. RIP3, the core regulatory protein of necroptosis, activates inflammasome 3 and caspase-1, thus promoting the secretion of the pro-inflammatory cytokines TNF-α and IL-1β [172]. Therefore, it may be a trigger for neuroinflammation [55,61,173,174,175] and promotes neuropathic pain via activated microglia [176]. A study on rat models of paclitaxel (PTX)-induced hyperalgesia reported that the necroptosis-related proteins RIP3/MLKL regulated neuronal necroptosis and increased the levels of pro-inflammatory cytokines in DRG [177]. In another study on rat models, peripheral nerve injury induced by CCI or SNI increased the expression of TNF-α, RIP1 and/or RIP3 [176,178] in the spinal cord, whereas Nec-1, an effective inhibitor of RIP1 and RIP-mediated necroptosis [65,159], significantly reduced the levels of spinal pro-inflammatory cytokines and RIP1/RIP3 and alleviated neuropathic hyperalgesia and allodynia [176,178].

Neuroinflammation owing to abnormally elevated TNF-α levels in the primary sensory afferent, spinal cord and ACC following peripheral nerve injury contributes to neuropathic pain [20,21,71]. Peri-sciatic administration of exogenous TNF-α without any nerve injury induces mechanical allodynia by activating the NF-kappaB pathway via an autocrine mechanism [71]. The TNF-α/TNFR1–necroptosis pathway may be a new and important target for research into chronic pain (Figure 2).

### 4.4. TNF-α/Necroptosis in Pain-Associated Anxiety and Depression

Chronic pain is often accompanied by adverse pain-associated emotional reactions such as anxiety and depression [179,180]. Several studies have been reported on emotional problems and chronic pain [181,182,183]. An increase in the levels of proinflammatory cytokines [184,185] and a decrease in the levels of neurotrophins [186,187] are related to emotional disorders. ACC, the first-order cortical region that responds to painful stimuli [188], plays an important role in pain information processing [189,190,191], including the processing of pain affection [192,193,194,195]. Abnormal expression of TNF-α, neuronal hyperexcitability and microglial activation in ACC contribute to inflammatory and neuropathic pain and pain aversion [20,196]. Studies have shown that pharmacologically blocking neuroinflammation and activation of glial cells of ACC reduces chronic pain and prevents the occurrence of accompanying emotional disorders or memory deficits caused by complete Freund’s adjuvant (CFA), the chemotherapeutic drug oxaliplatin or peripheral nerve injury [20,197,198].

Earlier studies have confirmed that in patients with severe depression, the loss of glial cells and reduction of neuronal size occur in the deeper cortical layers in ACC and the dorsolateral prefrontal cortex [199,200]. RIPK1, the key protein initiating RIPK3/MLKL-dependent necroptosis [201], promotes ischaemia-induced neuronal and astrocytic cell death [202]. A study reported that depression induced by chronic unpredictable mild stress (CUMS) led to anxiety-like behaviour but did not damage spatial learning and memory, which was accompanied by the expression of RIPK3/MLKL and activation of necroptosis [203]. Pharmacological or genetic regulation of necroptosis alleviates depressive or anxiety-like behaviour and improves hippocampal function and neuroinflammation [204]. Recently, the relationship between oligodendrocytes and emotional disorders has been receiving increasing attention [205,206,207,208,209,210]. In addition to the activation of microglia and impairment of astrocytic function, a reduction in the number or density of oligodendrocytes is one of the most prominent observations in depression [211]. Inflammatory cytokines from oligodendrocytes have been implicated in the pathological process of depression [212,213]. Inhibiting the activity of oligodendrocytes using a chemogenetic approach leads to depression-like behavior and increases TNF-*α*-induced oligodendrocyte necroptosis through interaction with TNFR1 [214]. Although studies reporting on the role of the TNF-α–necroptosis pathway in pain-associated mood disorders are lacking, research prospects of this pathway in the field of neuroscience can be predicted based on the above mentioned studies.

### 4.5. TNF-α/Necroptosis in Pain-Associated Memory Deficits

Depression is closely related to a decline in cognitive abilities such as concentration and memory difficulties [215] with a decreased volume of the hippocampal brain region [216]. These cognitive impairments are commonly related to chronic pain [217,218]. Decreased hippocampal volume can be observed in patients with chronic pathological pain such as chronic back pain, complex regional pain syndrome [219] and knee osteoarthritis [220]. Clinical studies have shown that chronic pain accompanied by a reduction in the hippocampal volume can significantly reduce the learning ability of the body [221,222], resulting in short- and long-term memory defects [223]. At present, the widely accepted theory for the mechanism of pain-impaired memory is the alteration of hippocampal synaptic plasticity (long-term potentiation [LTP]), which is considered the molecular mechanism underlying learning and memory [224,225] or that underlying the effects of pain on memory [117,219]. In addition, morphological and biochemical changes in the hippocampal region underlie cognitive impairment in neuropathic pain [226].

Upregulation of TNF-α [117,227,228], a decrease in the expression of brain-derived neurotrophic factor (BDNF) and microglial activation [226,227] in the hippocampus may be the basis of chronic pain and memory defects. Blocking nerve injury-induced hippocampal TNF-α via oral administration of magnesium L-threonine, a new method of preventing neuropathic pain caused by chemotherapy [229] or using nanocurcumin can improve pain and memory impairment [117,230]. At present, whether TNF-α mediates pain-associated cognitive deficits by activating necroptosis remains unclear; however, its role has been reported in other neurodegenerative diseases. For example, necroptosis mediates TNF-α-induced toxicity of hippocampal neurons [231], leading to memory impairment in AD [159,232,233]. Inhibiting necroptosis and abnormally high expression of TNF-α in the hippocampus can reduce cell death and improve cognitive ability [159,234,235,236].

## 5. Conclusions

The neuroimmune mechanisms underlying neuropathic pain are complex and involve many factors, including inflammatory and anti-inflammatory imbalances that in the most severe form is called cytokine storm. For example, spinal formyl peptide receptor type 2 (FPR2/ALX), a member of the formyl peptide receptors family, plays an analgesic role by reducing cytokines and BDNF [237]. This review focuses on the potential mechanisms of proinflammatory cytokine TNF-α-mediated neuropathic pain and discusses the role of TNF-α-necroptosis pathway in detail. As shown in Figure 2, TNF-α, as a crucial driver, can regulate cation channels to sensitize primary afferents in the peripheral nervous system, affect excitatory and inhibitory synaptic transmissions in CNS and evoke positive feedback between TNF-α and microglial activation to induce neuroinflammation, thus facilitating pain transmission, adverse pain-associated emotional reactions and cognitive deficits. TNF-α-triggered necroptosis, a novel form of programmed cell death, may be one of the key factors for inducing neuroimmune responses in neuropathic pain. It not only contributes to allodynia and hyperalgesia but also mediates aversion, anxiety, depression and learning and memory deficits associated with chronic pain. Therefore, understanding the important role of the TNF-α–necroptosis pathway in neuropathic pain may offer novel strategies for the treatment of neurological diseases.

## Figures and Tables

**Figure 1 ijms-23-07191-f001:**
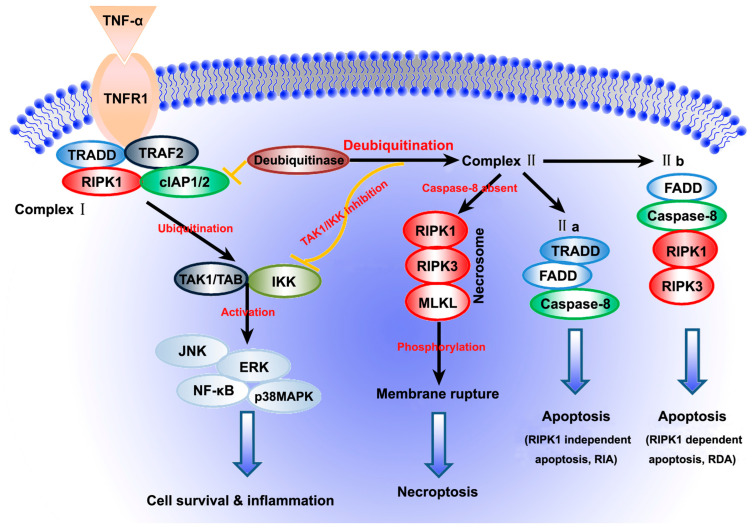
Tumor necrosis factor-alpha (TNF-α)/TNF receptor 1 (TNFR1)-mediated inflammatory response and cell death pathways. The binding of TNF-α to TNFR1 triggers inflammatory responses, apoptosis and necroptosis. Ubiquitination of receptor interacting protein kinase 1 (RIPK1) promotes cell survival and induces an inflammatory response by activating the NF-κB, p38 MAPK, JNK and ERK signalling pathways. If caspase-8 is present in cells, deubiquitination of RIPK1 results in the formation of either complex IIa or complex IIb, leading to RIPK1-independent apoptosis (RIA) or RIPK1-dependent apoptosis (RDA), respectively. If caspase-8 is absent, necroptosis is initiated, in which RIPk1, receptor-interacting protein kinase 3 (RIPK3) and mixed lineage kinase domain-like (MLKL) protein play a key role.

**Figure 2 ijms-23-07191-f002:**
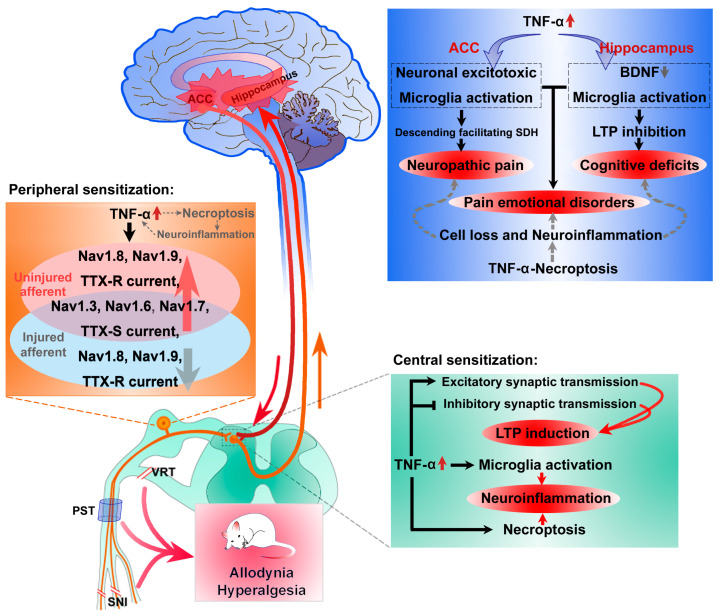
Potential mechanisms underlying peripheral and central sensitization via TNF-α or the TNF-α–necroptosis pathway in neuropathic pain. TNF-α regulates voltage-gated sodium channels (VGSCs) to sensitize primary afferents in the peripheral nervous system, affects excitatory and inhibitory synaptic transmissions in central nervous system (CNS), and evokes positive feedback between TNF-α and microglial activation to induce neuroinflammation, thus facilitating pain transmission, adverse pain-associated emotional reactions and cognitive deficits. TNF-α-triggered necroptosis in the dorsal root ganglia (DRG), spinal cord and supraspinal region may be one of the key factors for inducing neuroimmune responses in neuropathic pain.

**Figure 3 ijms-23-07191-f003:**
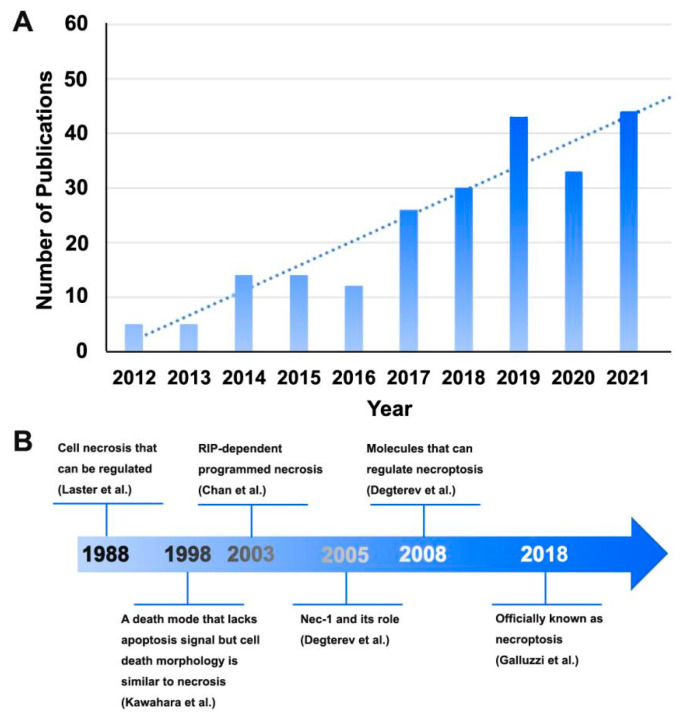
Global trends in the publication of necroptosis-related articles in the field of neuroscience analyzed via bibliometric analysis. (**A**) The number of articles published worldwide shows a steady upward trend. (**B**) The key nodes of necroptosis research [65,141,142,143,144,145].

**Figure 4 ijms-23-07191-f004:**
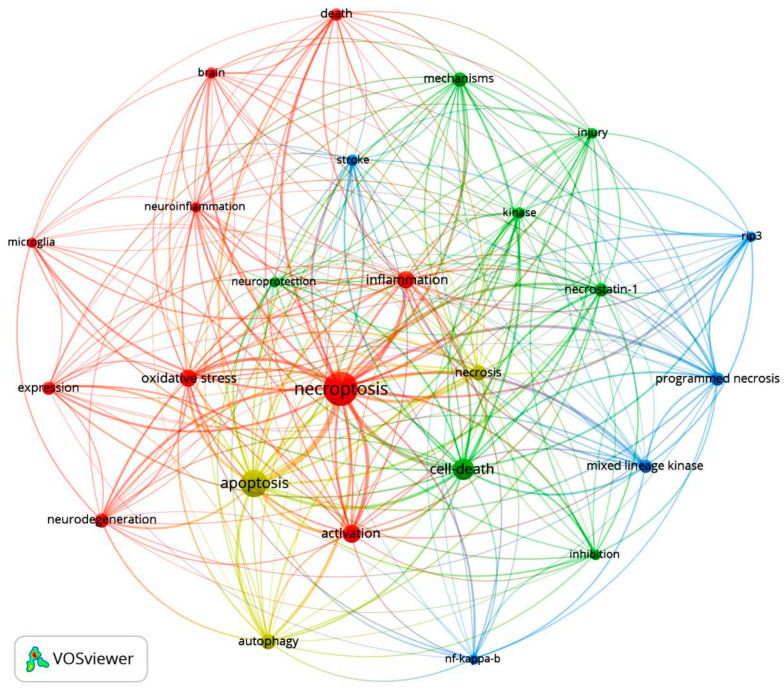
Co-occurrence network of keywords in necroptosis research established using the VOSviewer software. Four categories of necroptosis-related keywords are shown in red, blue, green and yellow. The red clusters are largest, and ‘necroptosis’ constitutes the largest node. Keywords associated with neuroinflammation and neurodegeneration such as ‘inflammation’, ‘oxidative stress’, ‘activation’ and ‘neurodegeneration’ are also mentioned.

**Table 1 ijms-23-07191-t001:** Top 10 journals with the highest co-citations in articles reported on necroptosis in the field of neuroscience.

Rank	Popular Journals	Co-Citations (n)	IF(2021)	Research Directions
1	Cell	341	41.584/Q1	Cell biology/stem cells
2	J biol chem	310	5.157/Q2	Signal transduction
3	J neurosci	275	6.167/Q1	Neuroscience/electrophysiology
4	P natlacad sci usa	274	11.205/Q1	Biology/physics
5	Cell death differ	256	15.828/Q1	Molecular biology/cell differentiation
6	Nature	244	49.962/Q1	Life sciences/natural science
7	J neurochem	206	5.372/Q1	Neuroinflammation/microglia
8	Brain res	187	3.252/Q3	Neuroscience/neuroprotection
9	Stroke	181	7.914/Q1	Stroke/cardio cerebrovascular diseases
10	Science	169	47.728/Q1	Catalysis/inheritance

**Table 2 ijms-23-07191-t002:** Top 10 necroptosis-related articles with the highest co-citations in the field of neuroscience.

Rank	Source	Citations (n)	Main Results	Research Directions	Ref.
1	Nat Chem Biol	74	Identification of necroptosis and its inhibitor Nec-1	New pathway of cell death	[65]
2	Nat Chem Biol	48	Necrostatinsarea family offirst-in-class inhibitors of RIP1 kinase, the key upstream kinase involved in the activation of necroptosis	Inhibition of necroptosis and its mechanism	[144]
3	Nat Rev Mol Cell Biol	43	Necroptosis can occur in a regulated manner	Molecular mechanisms of necroptosis	[151]
4	Cell	42	MLKL is a key mediator of necroptosis signalling downstream of RIP3 kinase	Molecular mechanisms of necroptosis	[152]
5	Cell	32	RIP3 controls programmed necroptosis by initiating the pronecrotic kinase cascade	Molecular mechanisms of necroptosis	[153]
6	Science	29	RIP3 is a molecular switch between TNF-induced apoptosis and necrosis and is required for RIP1-mediated necrosis	Molecular mechanisms of necroptosis	[154]
7	Cell	29	RIP3 as a determinant for cellular necrosis is recruited to RIPK1 to form a necrosis-inducing complex	Molecular mechanisms of necroptosis	[155]
8	Cell rep	28	Necroptosis is involved in multiple sclerosis (MS), and RIPK1 may represent a therapeutic strategy	Role of necroptosis in MS	[156]
9	Nature	28	A review of the regulatory mechanisms of necroptosis and its potential role in inflammation and diseases	Molecular mechanisms of necroptosis and its role in inflammation	[61]
10	Nat Immunol.	24	RIP is required for caspase-independent necrotic death induced by Pas, TNF and TRAIL	Molecular mechanisms of cell death	[54]

**Table 3 ijms-23-07191-t003:** Top 10 necroptosis-related articles with the highest citations in the field of neuroscience.

Rank	Source	Citations (n)	Main Results	Research Directions	Ref.
1	Neuron	244	Necroptosis drives motor neuron death in models of both sporadic and familial amyotrophic lateral sclerosis (ALS)	Role of necroptosis in ALS	[157]
2	Nat Rev Neurosci	197	Review of necroptosis in neurological diseases	Role of necroptosis in neurological diseases	[158]
3	Nat Neurosci	152	Genes regulated by RIPK1 overlap with multiple transcriptomic signatures of Alzheimer’s disease (AD)	Role of necroptosis in AD	[159]
4	Prog Neurobiol.	148	Review of the regulation of autophagy in neurons, glia, and brain microvascular cells in response to ischemia stress	Crosstalk between autophagy, necroptosis and apoptosis	[160]
5	J Pineal Res	136	Melatonin inhibits the Ripk3–PGAM5–CypD–mPTP cascade and hence reduces necroptosis	Role of necroptosis in ischaemia–reperfusion injury	[161]
6	J Neuroinflammation	131	Review of the molecular mechanisms of necroptosis and its relevance to diseases	The molecular mechanisms of necroptosis and its role in disease	[147]
7	Front Neurosci	131	Review of targeting Nrf2 to suppress ferroptosis and mitochondrial dysfunction in neurodegeneration	Neuroprotective signalling pathways	[162]
8	Adv Exp Med Biol	92	Review stating that the biochemical pathways causing programmed neurodegeneration, instead of neuronal death per se, are responsible for epileptogenesis	Reprogramming of neuronal death pathways in epileptogenesis	[163]
9	J Neurosci	88	The axodestructive factor Sarm1 is required for mitochondrial depolarisation-induced axon degeneration and cell death	A novel form of programmed cell destruction called sarmoptosis	[164]
10	Nat Neurosci	87	Efficient remyelination requires the death of microglia followed by their repopulation to a pro-regenerative state	Role of microglia in white matter regeneration	[53]

## Data Availability

Data were retrieved with the theme of “necroptosis” from the Division of Neuroscience in the core database of the Web of Science (WoS).

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
