# Peer review of "Neuroimmune Mechanisms Underlying Neuropathic Pain: The Potential Role of TNF-α-Necroptosis Pathway"

_ijms, 2022, doi:10.3390/ijms23137191_

Round 1

Reviewer 1 Report

The paper includes the latest data from the literature on the involvement of THF-α in the inflammatory response in apoptosis and necroptosis. the Role of TNF-α and its mechanism underlying neuropathic pain and TNF-α / TNFR1 – necroptosis pathway in neuropathic pain is presented coherently and clearly. 

This manuscript is very useful because the last article of this type is from 2010 (Leung L, Cahill CM. TNF-alpha and neuropathic pain - a review. J Neuroinflammation. 2010; 7:27. Published 2010 Apr 16. two: 10.1186 / 1742-2094-7-27). 

Author Response

Thank you very much for your comment.

Reviewer 2 Report

dear Authors

The study is interesting and deserves consideration in this journal, the topic is within the aim and scope and the text is correctly organized.

However I feel to suggest the following changes/modifications before publication:

1. English should be polished for typos

2. Figures resolution should be improved

3. mechanism of inflammation are always complex and several factors should be considered when talking about neuropathic pain. I suggest to consider the correlation between it and FPR system in CNS, as is describe dby several authors, for example: "New Insights on Formyl Peptide Receptor Type 2 Involvement in Nociceptive Processes in the Spinal Cord", "Role of formyl peptide receptors (FPR) in abnormal inflammation responses involved in neurodegenerative diseases", "Effects of kisspeptin-10 on hypothalamic neuropeptides and neurotransmitters involved in appetite control".

Author Response

Point 1: English should be polished for typos

Response 1: Thank you. The manuscript has been revised thoroughly. All changes in the manuscript are marked with RED.

Point 2: Figures resolution should be improved

Response 2: Thank you for your suggestion. We confirmed again that all the pictures in the article have a resolution of 300 dpi or above.

Point 3: mechanism of inflammation are always complex and several factors should be considered when talking about neuropathic pain. I suggest to consider the correlation between it and FPR system in CNS, as is describe dby several authors, for example: "New Insights on Formyl Peptide Receptor Type 2 Involvement in Nociceptive Processes in the Spinal Cord", "Role of formyl peptide receptors (FPR) in abnormal inflammation responses involved in neurodegenerative diseases", "Effects of kisspeptin-10 on hypothalamic neuropeptides and neurotransmitters involved in appetite control".

Response 3: Thank you very much for your comment. This paragraph “The neuroimmune mechanisms underlying neuropathic pain are complex and involve many factors, including inflammatory and anti-inflammatory imbalances. For example, in the identification of pathogen-associated molecular patterns and damage-associated molecular patterns, formyl peptide receptors (FPRs) play a vital role in the innate immune response [267]. Spinal formyl peptide receptor type 2 (FPR2/ALX), a member of the FPRs family, plays an analgesic role by reducing cytokines and BDNF [268]. This review focuses on the potential mechanisms of proinflammatory cytokine TNF-α-mediated neuropathic pain, and discusses the role of TNF-α-necroptosis pathway in detail. ” was added to lines 385 to 394 (red) and two references, numbered 267 and 268, have been added to lines 1015 to 1018 (red).

Reviewer 3 Report

An interesting and important review. I note, as a minor concern, that the conclusions section is too concise according to the data of the review. I suggest that this section could be lenghtened. 

Author Response

Point 1: An interesting and important review. I note, as a minor concern, that the conclusions section is too concise according to the data of the review. I suggest that this section could be lenghtened. 

Response 1: Thank you very much for your comment. This paragraph “The neuroimmune mechanisms underlying neuropathic pain are complex and involve many factors, including inflammatory and anti-inflammatory imbalances. For example, in the identification of pathogen-associated molecular patterns and damage-associated molecular patterns, formyl peptide receptors (FPRs) play a vital role in the innate immune response [267]. Spinal formyl peptide receptor type 2 (FPR2/ALX), a member of the FPRs family, plays an analgesic role by reducing cytokines and BDNF [268]. This review focuses on the potential mechanisms of proinflammatory cytokine TNF-α-mediated neuropathic pain, and discusses the role of TNF-α-necroptosis pathway in detail. ” was added to lines 385 to 394 (red) and two references, numbered 267 and 268, have been added to lines 1015 to 1018 (red).